# Allopatric Lineage Divergence of the East Asian Endemic Herb *Conandron ramondioides* Inferred from Low-Copy Nuclear and Plastid Markers

**DOI:** 10.3390/ijms232314932

**Published:** 2022-11-29

**Authors:** Kuan-Ting Hsin, Hao-Chih Kuo, Goro Kokubugata, Michael Möller, Chun-Neng Wang, Yi-Sheng Cheng

**Affiliations:** 1Department of Life Science, College of Life Science, National Taiwan University, Taipei 10617, Taiwan; 2Biodiversity Research Center, Academia Sinica, Taipei 11529, Taiwan; 3Department of Botany, National Museum of Nature and Science, Tsukuba 305-0005, Japan; 4Royal Botanic Garden Edinburgh, Edinburgh EH3 5LR, UK; 5Institute of Plant Biology, College of Life Science, National Taiwan University, Taipei 10617, Taiwan; 6Genome and Systems Biology Degree Program, College of Life Science, National Taiwan University, Taipei 10617, Taiwan

**Keywords:** *Conandron*, least-cost path, ornamental plant, single-copy nuclear marker, vicariance

## Abstract

The evolutionary histories of ornamental plants have been receiving only limited attention. We examined the origin and divergence processes of an East Asian endemic ornamental plant, *Conandron ramondioides*. *C. ramondioides* is an understory herb occurring in primary forests, which has been grouped into two varieties. We reconstructed the evolutionary and population demography history of *C. ramondioides* to infer its divergence process. Nuclear and chloroplast DNA sequences were obtained from 21 *Conandron* populations on both sides of the East China Sea (ECS) to explore its genetic diversity, structure, and population differentiation. Interestingly, the reconstructed phylogeny indicated that the populations should be classified into three clades corresponding to geographical regions: the Japan (Honshu+Shikoku) clade, the Taiwan–Iriomote clade, and the Southeast China clade. Lineage divergence between the Japan clade and the Taiwan–Iriomote and Southeast China clades occured 1.14 MYA (95% HPD: 0.82–3.86), followed by divergence between the Taiwan–Iriomote and Southeast China clades approximately 0.75 MYA (95% HPD: 0.45–1.3). Furthermore, corolla traits (floral lobe length to tube length ratios) correlated with geographical distributions. Moreover, restricted gene flow was detected among clades. Lastly, the lack of potential dispersal routes across an exposed ECS seafloor during the last glacial maximum suggests that migration among the *Conandron* clades was unlikely. In summary, the extant *Conandron* exhibits a disjunct distribution pattern as a result of vicariance rather than long-distance dispersal. We propose that allopatric divergence has occurred in *C. ramondioides* since the Pleistocene. Our findings highlight the critical influence of species’ biological characteristics on shaping lineage diversification of East Asian relic herb species during climate oscillations since the Quaternary.

## 1. Introduction

Ornamental plants are known for their aesthetic appeal, and their horticultural history can be traced back to ancient Egypt [1]. At least 4500 ornamental plants originate from thirteen geographic regions worldwide, one of which is East Asia [2]. East Asia is known for its high plant diversity and endemism [3]. Landscape configuration changes caused by climatic oscillation and abundant landscape diversity are two key components in shaping plant diversity and endemism in the region [4,5].

It has been proposed that population differentiation is the driving force for plant speciation, which should involve the process of how barriers affect gene flow between previously interbreeding species [6,7]. East Asia comprises the Asia continent and adjacent islands, including the Japan archipelagos, the Ryukyu archipelagos, and the island of Taiwan. Since the Quaternary, these (currently separated) landmasses were repeatedly connected during glacial–interglacial periods [8,9], suggesting the appearance and disappearance of geographical barriers in this region through time. Two models are used to explain how species exhibit the disjunct distribution patterns often found under such circumstances. The first involves long-distance dispersal among suitable habitats. The second invokes vicariance owing to a range contraction of previously widely distributed populations. Relatively little is known regarding gene flow changes among separated populations after long-distance dispersal or vicariance has influenced population differentiation.

In East Asia, many plant species exhibit continent–island distribution patterns [10,11,12,13,14,15]. Studies of the genetic compositions of *Cercidiphyllum japonicum* [11], *Kalopanax septemlobus* [13], *Quercus variabilis* [10], and *Viola orientalis* [14] showed a genetic mixture between the East Asia continent and adjacent island populations. In contrast, studies of *Platycrater arguta* [12] and *Dendrobium moniliforme* [15] found that haplotypes were not shared and gene flow was restricted between continent and island populations. Moreover, the lack of predicted suitable habitats on the exposed East China Sea (ECS) seafloor suggests that migration of *P. arguta* populations over land was unlikely [12]. In studying the lineage divergence process of *P. arguta*, estimated restricted gene flow between populations distributed on the continent (China) and on islands (Japan) suggests a lack of migration between these populations since their first divergence. Exploring genetic compositions and modeling suitable habitat ranges from past geographical periods can allow researchers to explore the divergence process of target species. However, the role of post-divergence gene flow between continent- and island-distributed populations in shaping lineage divergence has not been sufficiently examined among the species mentioned above.

*Conandron ramondioides* is not only an East Asian endemic temperate understory herb but also an ornamental plant in the plant market. *C. ramondioides* is distributed across East China, Taiwan, southern Okinawa, and Japan but is absent in between these locations, thus exhibiting a disjunct distribution pattern. This has been interpreted as the species having migrated from the Asian continent to adjacent islands or fragmented from an ancient mega population in East Asia. Based on morphological traits (lobe length to tube length ratio) of samples obtained from Japan and Taiwan, *C. ramondioides* has previously been assigned to two varieties: *C. ramondioides* var. *ramondioides* and *C. ramondioides* var. *taiwanensis* [16]. However, the lineage divergence mechanism of *C. ramondioides* is not yet known. Two scenarios have been proposed so far: one is a recent long-distance dispersal scenario, and the other is a vicariance scenario. In the former, island-distributed *C. ramondioides* populations may have originated from the Asian continent, followed by dispersal to adjacent islands. The vicariance scenario proposes that populations currently distributed on the Asian continent and islands are fragments of an ancient mega population. We were especially interested in whether gene flow between *C. ramondioides* var. *ramondioides* and *C. ramondioides* var. *taiwanensis* took place during their evolutionary history. We used molecular markers and phylogeographical analysis to verify which scenario is the most likely to have occurred.

In phylogeographical studies, chloroplast DNA sequences (cpDNA) have been used successfully to infer relationships among closely related species or populations of plants [17,18,19,20]. However, cpDNA, like any other single gene (e.g., *LEAFY*), is subject to stochastic processes. The results of single-gene trees may be over-interpreted if researchers do not consider coalescent stochasticity [21]. To prevent over-interpretation of genealogy from a single gene, one can combine multiple genes (e.g., nuclear DNA loci) to provide more reliable inferences of population colonization history and population dynamics through time. Applying multiple nuclear DNA loci to infer population colonization history and population dynamics through time is advantageous because nuclear DNA is inherited from two parents, and the substitution rates are often faster in nuclear loci than those of molecular markers in the cpDNA [22]. Using both cpDNA and nrDNA markers has the combined advantage of enabling the identification of evolutionary history differences between organelles and populations.

In this study, we performed a phylogeographic survey of *C. ramondioides* by examining genetic variations obtained from both nrDNA (including *ATG2* intron 1, *GroES* intron 1, *LEAFY* intron 1, *ITS,* and *CrCYC1*) and cpDNA (including *trn*L-*trn*F and *trn*H-*psb*A) markers of 21 *C. ramondioides* populations in East Asia. We aimed (i) to reveal genetic relationships, approximate divergence time, and gene flow between continent- and island-distributed *Conandron* populations; (ii) to verify whether their taxonomic classification corresponds to their genetic relationships; and (iii) to reveal the potential suitable habitat range via simulation of *Conandron* during the last glacial maximum (LGM).

## 2. Results

### 2.1. Haplotype Networks Reconstructed from Four Low-Copy Markers Reveal Three Groups of C. ramondioides Haplotypes Which Correspond to Geographical Locations

To infer the phylogeographic distribution pattern of *C. ramondioides* in East Asia (Figure 1), we reconstructed the haplotype network of each molecular marker used in this study. In total, 38, 47, 28, and 45 haplotypes were identified from the ATG intron 1, GroES intron 1, LFY intron 1, and *CrCYC1* datasets, respectively (Figure 2A–D). Among the reconstructed haplotype network of the ATG intron 1, GroES intron 1, and LFY intron 1 datasets, the haplotypes distributed across SE-China formed a single group (Figure 2A–C). Haplotypes occurring in Taiwan and Iriomote descended from one Taiwan–Iriomote shared haplotypes among these three reconstructed haplotype networks, while haplotypes distributed in Honshu and Shikoku formed multiple lineages (Figure 2A–C). The haplotypes distributed across SE-China were two to four mutational steps away from the haplotypes distributed in the Taiwan–Iriomote region, while those distributed in Honshu and Shikoku were two to seven mutational steps away (Figure 2A–C). In the reconstructed *CrCYC1* haplotype network, haplotypes formed three distinct groups, which corresponded to their geographical region with 10 to 11 mutational steps between each group (Figure 2D).

Among these four haplotype networks, the *CrCYC1* topology formed three distinct groups with long mutational steps between clades. The phenomenon might be due to either the fast evolution of *GCYC* in Gesneriaceae or the fact that *GCYC* has been subjected to non-neutral evolution, or both. Furthermore, *GCYC* is a functional gene in regulating floral symmetry in Gesneriaceae and is subjected to selection. *CrCYC1* is a member of the *GCYC*. To evaluate whether *CrCYC1* evolved neutrally, the McDonald–Kreitman test was performed (see “Significant MK test results suggested that *CrCYC1* deviates from the neutral evolution hypothesis” section). In summary, haplotype networks reconstructed from four low-copy nuclear markers suggested that haplotypes of *C. ramondioides* formed three groups corresponding to geographical regions.

### 2.2. Two Groups of C. ramondioides Are Proposed from the Reconstructed ITS Ribotype Network

Two groups may be identified from the reconstructed *ITS* ribotype network. One group includes haplotypes distributed across Honshu, SE-China, Taiwan, and Iriomote, with rare ribotypes connecting to it. The second group represents ribotypes descending from ribotypes distributed in Honshu–Iriomote, which differ from the former ribotype by four mutational steps (Figure 2E).

Unlike the three haplotype groups reconstructed from low-copy nuclear markers, noted above, multiple haplotypes connected to one geographically broadly distributed *ITS* ribotype were situated in the center of the *ITS* ribotype network (Figure 2E). The *ITS* marker is a product of concerted evolution, a process leading to the homogenization of different genes in a gene family—our sequencing chromatograms show multiple peaks at one site. Since *C. ramondioides* is reported as a diploid species, we expected no more than two types of alleles of *C. ramondioides ITS* to be present. Clone technology was applied to determine whether *C. ramondioides ITS* achieves concerted evolution.

### 2.3. Two Groups of C. ramondioides Are Shown by the Reconstructed cpDNA Haplotype Network

In the reconstructed cpDNA haplotype network, multiple lineages connected to a SE-China- and Taiwan-distributed haplotype form a star-like topology. Honshu- and Shikoku-distributed haplotypes formed two single lineages in the reconstructed haplotype network (Figure 2F).

When comparing the cpDNA topology with the other topologies, the Southeast China group cpDNA haplotypes formed multiple lineages instead of the monotypic lineages shown by the other four low-copy nuclear markers. As cpDNA is a maternally inherited and haploid genome, genetic drift should influence it at least four times faster than nrDNA, which should result in more complete lineage sorting. However, this was not the case in the China cpDNA haplotype. Therefore, both Tajima’s D and Fu’s Fs tests were run to examine whether cpDNA isolated from *Conandron* occurring in China deviates from neutral evolution.

### 2.4. Significant MK Test Results Suggest That CrCYC1 Deviates from the Neutral Evolution Hypothesis

To evaluate whether the *CrCYC1* gene deviates from the neutral evolution hypothesis, a McDonald–Kreitman (MK) test was run. If Pn/Ps is not significantly different from Dn/Ds, this indicates that *CrCYC1* does not deviate from the neutral hypothesis. The results show that Pn/Ps within *C. ramondioides* (20/5) was significantly higher than Dn/Ds between *C. ramondioides* and *C. umbellifera* (24/24; *p* = 0.013) (Table 1). The higher within-species Pn/Ps ratio than between-species Dn/Ds ratio was also observed when using *GCYC1* of *Hemiboea bicornuta* and *Oreocharis benthamii* as outgroup sequences, respectively (Table 1). Significant MK test results suggest that the *CrCYC1* dataset deviated from the neutral hypothesis, and it was thus excluded from further analysis.

### 2.5. Multiple ITS Haplotypes Were Identified from Selected Conandron Individuals

Eight *Conandron* individuals (two from Honshu and Shikoku, three from Southeast China, and three from Taiwan) were used to verify whether internal transcribed spacers (*ITS*) amplified from *Conandron* were subjected to concerted evolution. We obtained multiple haplotypes (ranging from three to six) from colony sequences (Appendix A). High haplotype numbers both suggested high intra-individual polymorphism of *Conandron ITS* and indicate incomplete concerted evolution of *Conandron ITS*. In addition, a recombination event is also detected in this dataset. Both incomplete concerted evolution and recombination contribute to *ITS* diversity in *Conandron*. Therefore, we excluded *ITS* from our multi-nuclear loci dataset to prevent it from blurring evolutionary signals of the *Conandron* populations.

### 2.6. Significant Tajima’s D and Fu’s Fs Suggest cpDNA Amplified from Southeast China Deviate from Neutral Evolution

The neutral theory assumes that a small effective population size is more likely to be significantly affected by evolutionary processes than a large effective population size. In diploid species, the effective population size of nrDNA is four times greater than that of cpDNA, and consequently, the effect of genetic drift should be four times faster. However, our results are contrary to this expectation. In the haplotype networks, China haplotypes formed a monophyletic group based on the three single-copy nuclear markers (Figure 2A–C) but not on the cpDNA marker (Figure 2F). We therefore assumed that the cpDNA from China may not have evolved neutrally. To test this assumption, we calculated Tajima’s D and Fu’s Fs to examine deviation from neutrality. We applied these two neutrality tests to both nrDNA and cpDNA markers.

Our results show that both test metrics significantly departed from neutrality in the cpDNA dataset, in contrast to those from the nrDNA markers (Table 2). The positive value derived from the neutrality test suggests that China’s population may experience diversifying selection. This result may explain why the China populations of *C. ramondioides* maintain genetic variation instead of losing genetic variation via genetic drift.

### 2.7. Genetic Diversity Measured in Conandron Populations

In total, 208 AGT intron 1 sequences, 270 GroES intron 1 sequences, and 192 LFY intron 1 sequences were retrieved from our ampliqon library. Reads of AGT intron 1 ranged from 102 reads to 1524 reads, reads of GroES intron 1 ranged from 524 reads to 3383 reads, and reads of LFY intron 1 ranged from 102 reads to 3011 reads. Comparisons of the genetic diversity indices (Hd and π) calculated from these molecular markers of *C. ramondioides* populations adjacent to the ECS provided equivocal genetic diversity index patterns (North: Hd, π vs. South: Hd, π, North_ATG2 intron1_: Hd = 0.736; π = 4.64 × 10^−3^ vs. South_ATG2 intron1_: Hd = 0.9; π = 5.64 × 10^−3^, North_GroES intron1_: Hd = 0.925; π = 13 × 10^−3^ vs. South_GroES intron1_: Hd = 0.818; π = 4.71 × 10^−3^, North_LEAFY intron1_: Hd = 0.823; π = 6.63 × 10^−3^ vs. South_LEAFY intron1_: Hd = 0.818; π = 6.15 × 10^−3^, North_CrCYC1_: Hd = 0.897; π = 4.7 × 10^−3^ vs. South_CrCYC1_: Hd = 0.937; π = 10.32 × 10^−3^, North_ITS_: Hd = 0.802; π = 4.43 × 10^−3^ vs. South_ITS_: Hd = 0.839; π = 3.43 × 10^−3^, North_cpDNA_: Hd = 0.633; π = 2.49 × 10^−3^ vs. South_cpDNA_: Hd = 0.894; π = 3.95 × 10^−3^) (Figure 3A,B; details are listed in Appendix A). A comparison of these two genetic indices obtained from populations located on either side of the Taiwan strait also showed equivocal patterns (TI: Hd, π vs. C: Hd, π, TI_ATG2 intron1_: Hd = 0.86; π = 5.2 × 10^−3^ vs. C_ATG2 intron1_: Hd = 0.68; π = 2.16 × 10^−3^, TI_GroES intron1_: Hd = 0.659; π = 2.26 × 10^−3^ vs. C_GroES intron1_: Hd = 0.726; π = 4.19 × 10^−3^, TI_LEAFY intron1_: Hd = 0.738; π = 2.67 × 10^−3^ vs. C_LEAFY intron1_: Hd = 0.623; π = 5.02 × 10^−3^, TI_CrCYC1_: Hd = 0.911; π = 3.02 × 10^−3^ vs. C_CrCYC1_: Hd = 0.818; π = 5.54 × 10^−3^, TI_ITS_: Hd = 0.731; π = 3.45 × 10^−3^ vs. C_ITS_: Hd = 0.777; π = 3.93 × 10^−3^, TI_cpDNA_: Hd = 0.883; π = 4.47 × 10^−3^ vs. C_cpDNA_: Hd = 0.821; π = 6.34 × 10^−3^) (Figure 3C,D, details are listed in Appendix A).

### 2.8. Significant Fst Suggests Population Differentiation among Assigned Geographical Regions

Significant pairwise Fst values are measured among three assigned groups of all markers (Table 3). For the ATG2 intron 1, the measured Fst values ranged from 0.571 to 0.748. For the GroES intron 1, the measured Fst values ranged from 0.298 to 0.513. As to the LFY intron 1 dataset, the measured Fst values ranged from 0.472 to 0.848. Lastly, the measured Fst value of cpDNA ranged from 0.269 to 0.35. The hierarchical AMOVA analysis of nuclear markers showed that either the highest or second highest genetic variation was assigned to the “among group” class (Table 4). The values of genetic variation assigned to the “among group” class were 39.61%, 36.75% and 47.81% of ATG2 intron 1, GroES intron 1, and LFY intron 1, respectively.

### 2.9. Lineage Divergence Order and Divergence Time among C. ramondioides Lineages Revealed by nrDNA

To trace the lineage divergence process of *C. ramondioides*, the species phylogeny was reconstructed by using the genetic information from the aforementioned three nuclear markers, including ATG2 intron 1, GroEs intron 1, and LFY intron 1. The reconstructed phylogeny showed *C. ramondioides* was assigned into five monophyletic clades with moderately to highly statistic support (posterior probability ≥0.75) (Figure 4). The Taiwan- and Iriomote-distributed populations were grouped with high statistical support (posterior probability = 0.98). This clade is referred to as the Taiwan–Iriomote clade for further isolation with migration model analysis. Then, the Taiwan–Iriomote clade was grouped with populations distributed in Southeast China (posterior probability = 0.75). The Honshu- and Shikoku-distributed *C. ramondioides* populations formed a monophyletic clade with high statistic support (posterior probability = 1). Lastly, the estimated lineage divergence time between the Honshu–Shikoku clade and the Southeast China–Taiwan–Iriomote clades was dated back to 1.13 (95% CI: 0.26–1.8) million years ago (MYA), while the estimated divergence time between the Southeast-China-distributed population and those distributed in Taiwan and Iriomote occurred 0.75 million years ago (95% CI: 0.37–1).

For the multi-population IMa2 analysis, we present summary statistics in Figure 5 and the marginal posterior density distribution in Appendix A. The mean estimates for times of split indicate divergence between the SE-China lineage (C) and the Taiwan–Iriomote lineage (TI) (t0) and between the common ancestor (A) of the C, TI, and Honshu + Shikoku (HS) lineages (t1) occurring 0.75 MYA (95%HPD: 0.45–1.3) and 1.14 MYA (95%HPD: 0.82–3.86), respectively. However, the confidence interval of t1 was relatively broad and provided a flat marginal distribution (Appendix A). One of eight migration parameters (2Nm) had non-zero peaks between the common ancestor of C and TI to HS (Appendix A). Mean estimates of adequate population size for the extant lineages were 547,070 individuals for the HS group, 810,240 individuals for the TI lineage, and 122,784 individuals for the C group. Flat and broad marginal posterior density distributions were provided for the effective ancestral population sizes, which provides little information regarding these parameters (Appendix A).

To further evaluate the reliability of multi-population IMa2 analysis, a pairwise IMa2 analysis was conducted. The divergence event between the TI and C group occurred the most recently (ca. 0.67 MYA with 95%HPD: 0.28–0.7 MYA) (Figure 6). The lineage divergence times estimated between the HS–C and the HS–TI pairs are 1.11 and 0.97 million years ago, respectively (Figure 6). By arranging the divergence time estimated from our pairwise IMa2 analysis, one can conclude that the divergence between HS and the TI–C occurred firstly, following divergence between TI and C group. The highest estimated effective population size of assigned *C. ramondioides* was the TI group, then HS group and finally C group (Appendix A). The estimated geneflow among *C. ramondioides* groups all peaked at 0 (Appendix A). Lastly, the broad and flat marginal posterior probability curves of the ancestral population suggested little information available for those estimated ancestral population parameters.

### 2.10. Population Dynamic through Time of Conandron

Most sampled populations in Taiwan exhibit non-significant negative Tajima’s D values estimated from nuclear markers and cpDNA markers (see Appendix A). Non-significant Tajima’s D values were also observed in individual populations and regional groups in Honshu and Shikoku. The non-significant Tajima’s D values were also obtained from Southeast China populations. In contrast, significant negative Fu’s Fs values were observed for the regional dataset, including Taiwan and Taiwan–Iriomote–Southeast China (see Appendix A, *p* < 0.05, bold numbers). It is regarded that Fu’s Fs is more sensitive for identifying population expansion and past population growth. Therefore, population expansion may have been restricted to Taiwan and Iriomote. Furthermore, population contraction may have been restricted to individual Honshu and Shikoku populations due to most populations and regional groups exhibiting non-significant positive Fu’s Fs values (see Appendix A). Most individual population and regional groups showed non-significant or slightly negative or positive Tajima’s D and Fu’s Fs values, suggesting a lack of population dynamics of *Conandron* in East Asia.

To trace population size change through time, an EBSP was applied to the HS, C, and TI groups. Our EBSP reconstructed from the HS group showed that the HS group has remained stable in population size since 0.6 MYA, and then experienced a recent population size decline (Figure 7A, BF = 3.39). The EBSP of the Southeast China group exhibited flat and narrow credible intervals, which hint at the Southeast China group maintaining stable population size since 0.2 MYA (Figure 7B, BF = 2.23). Lastly, the EBSP curve of the TI group exhibit a gradually increasing trend since 0.2MYA (Figure 7C, BF = 509.28). To conclude, the *Conandron* groups maintained a stable effective population size trend in East Asia.

### 2.11. Modeled Migration Route of C. ramondioides during the LGM (Least-Cost Path)

The modeled potential suitable habitat ranges of *Conandron* are mainly located on current land masses (e.g., Taiwan, Honshu, and Southeast China) (Figure 8). Despite the fact that the ECS floor was exposed during the LGM, the modeled suitable habitat range for *Conandron* is restricted on the exposed ECS seafloor. Hence, no potential dispersal routes crossing the ECS seafloor were observed during the LGM. For *Conandron* populations, migration occurs within each landmass (e.g., within Japan, Southeast China, and Taiwan) (Figure 8).

### 2.12. Morphological Variations Measured in C. ramondioides Individuals Distributed on the Continent and Islands

Three morphological groups were distinguished by plotting averaged lobe length versus averaged tube length (L/T) of available *C. ramondioides* flowers (Figure 9). The L/T of individuals distributed in Taiwan ranged from 1.853 to 3.105; for those distributed in Japan, 1.053 to 1.575; and for those distributed in China, 0.339 to 0.37.

## 3. Discussion

### 3.1. High Genetic Diversity and Strong Genetic Differentiation of C. ramondioides

Although the *Conandron* populations are isolated from each other and disjunctively distributed in East Asia, *Conandron* shows similar or even higher levels of nuclear and cpDNA haplotype and nucleotide diversity (Figure 3 and Appendix A) than has been reported for similar species occurring in East Asia (e.g., Hd = 0.8862, π = 0.00361 of *Dendronbium*, Hd = 0.895, π = 0.00306 of *Ligularia*, and Hd = 0.8, π = 0.00385 of *Kirengeshoma*) [15,23,24]. A high level of genetic diversity is generally thought to reflect long evolutionary histories [25] but also suggests that inter-population gene flow is restricted, thereby allowing for maintenance of a high level of genetic diversity. Bumblebees, whose foraging distance is restricted (ca. 267 m) [26], are thought to be the major pollinators of *Conandron* in Taiwan [27]. This suggests that long-distance dispersal by pollen is unlikely for *Conandron*. Furthermore, seeds of *Conandron* lack both hair-like appendages for attachment to seed dispersers and any structures attractive to animal seed consumers. These two morphological characteristics of *Conandron* seeds suggest that long-distance seed dispersal is unlikely. Moreover, the short stature of *Conandron* and its understory forest habitat may discourage visits from insect pollinators [28], reducing interpopulation gene flow and facilitating inter-population differentiation. Significant pairwise Fst values between groups (Table 3) and the among-group ΦCT index (Table 4) support this assumption. Considering its reproductive (limited pollen dispersal by pollinators) and morphological (lack of seed-dispersal adaptations) characteristics, gene flow among extant *Conandron* populations via pollen and seeds is probably severely constrained. This pattern may be a consequence of the species’ requirements of a cool and moist environment in temperate deciduous forest habitats at mid-elevation. Due to its biological features (high sensitivity to environmental change, limited pollen dispersal by pollinators, and lack of long-distance seed-dispersal ability), the repeated exposure of the ECS seafloor may not have served as a land bridge allowing migration among *Conandron* populations since the Quaternary. To further examine the lineage divergence and identify whether post-divergence gene flow ever occurred, construction of a phylogeny and estimated post-divergence gene flow of *Conandron* were conducted (see below).

### 3.2. Middle Pleistocene Lineage Diversification of Conandron

The haplotype network analysis using single-copy nuclear markers showed that *Conandron* populations could be assigned to three groups corresponding to geographical regions: the Honshu + Shikoku group, the Southeast China group, and the Taiwan + Iriomote group (Figure 2A–C). This suggests that fragmentation is likely. This diversification process was also supported by a multi-loci phylogeny (Figure 4). In this phylogeny, extant Japanese *Conandron* populations diverged from those in Southeast China, Taiwan, and Iriomote, following divergence between the Southeast China and the Taiwan and Iriomote populations (Figure 4). The genetic indices (Hd and π) showed equivocal patterns among molecular markers (Figure 3A,B). The equivocal patterns of the genetic indices suggest that the currently observed dispersal scenario, either from north to south or from south to north, is unlikely. Furthermore, genetic indices measured from molecular datasets of the TI and C groups exhibit equivocal patterns (Figure 3C,D), suggesting that recent dispersal between the TI and C populations is unlikely.

The scaled divergence time estimated by IMa analysis showed that the first diversification event occurred at c. 1.14 (95% HPD, 0.82–3.86) MYA between the north (Honshu and Shikoku) and the south (Southeast China, Taiwan, and Iriomote) groups, while a second diversification event occurred at c. 0.75 (95% HPD, 0.45–1.3) MYA between the Southeast China and Taiwan–Iriomote groups (Figure 5 and Figure 6). Our estimate of divergence time among *Conandron* groups corresponds to the Middle to Late Pleistocene eras, in which at least four glacial–interglacial cycles occurred [29]. Thus, the genetic divergence and distribution range of *Conandron* has been associated with these glacial–interglacial cycles. Although the *Conandron* populations are now separated, with some located on the Asian continent and others on adjacent islands (Honshu, Shikoku, Taiwan, and Iriomote), these localities were once connected by an exposed ECS seafloor during glacial periods, potentially allowing *Conandron* to disperse into East Asia. However, restricted post-divergence gene flow was measured between the north and south groups and between the C (Southeast China) and TI (Taiwan and Iriomote) groups, suggesting that dispersal among *Conandron* populations since lineage divergence was unlikely (Figure 5 and Figure 6). Furthermore, the EBSP-reconstructed population size dynamic showed that *Conandron* populations either maintained a stable population size (Honshu and Southeast China; Figure 7A,B, respectively) or increased slightly (Taiwan and Iriomote; Figure 7C). Maintaining a stable or mildly increasing population size may indicate that there are limited suitable habitats for *Conandron* in these regions.

Reconstructed paleo-environment conditions on the ECS land bridge were colder and drier than at present [30,31,32]. Given that *Conandron* is a cold-tolerant but drought-intolerant understory herb, the relatively drier paleo-environment conditions on the ECS land bridge (compared with the present) were not favorable for *C. ramondioides*. The ENM-modeled suitable habitat range may provide further support for this hypothesis (Figure 8). The modeled suitable habitat ranges of *C. ramondioides* show a lack of suitable habitats on the ECS land bridge during the LGM (Figure 8), suggesting that migration of *C. ramondioides* over the land bridge was unlikely. To conclude, we identified a lack of geographically shared haplotypes, a restricted gene-flow post-divergence, and limited suitable habitat on the exposed ECS seafloor. This leads us to postulate that the extant disjunctly distributed *Conandron* experienced at least two vicariance events since the Middle Pleistocene.

Contrary to earlier hypotheses, the exposed ECS floor did not necessarily serve as a land bridge enabling migration of currently isolated populations or species. There is an increasing number of studies suggesting that the ECS land bridge acted as a barrier to some East Asian temperate plants by selectively hampering migration but allowing free migration of some other species. For example, genetic breaks across the ECS have been detected in the populations of several understory fern, herbs, orchids, and shrubs from temperate forests in East Asia (e.g., genus *Lemmaphyllum*, *Ligularia hodgsonii*, *Dendrobium moniliforme*, *Platycrater arguta*, *Euptelea pleiosperma,* and *Euptelea polyandra*) [15,23,24,33,34]. In contrast, genetic breaks were not identified in populations from both sides of the ECS in three widespread temperate forest tree species, *Quercus variabilis* [10], *Cercidiphyllum japonicum* [11], and *Kalopanax septemlobus* [13]. Reproductive characteristics of these trees, such as wind-dispersed pollen or seeds and vegetative propagation, would have made migration across the ECS land bridge during the glacial period possible. However, *Conandron* and other studied herbaceous species do not possess such reproductive traits. Thus, predictions regarding the role of the ECS land bridge have to take into account both the presence of the land bridge and other relevant biological features of each species, especially its reproductive characteristics.

### 3.3. Cryptic Diversification within C. ramondioides var. Taiwanensis

Based on its floral trait (lobe length to tube length ratio), two varieties were defined within the *C. ramondioides* [16]. One is *C. ramondioides* var. *ramondioides* (distributed in mainland Japan); the other is *C. ramondioides* var. *taiwanensis* (distributed in Taiwan and Iriomote). However, the Southeast-China-distributed *Conandron* is not included in the previous study. Here, our multi-loci phylogeny not only provides molecular support for the taxonomic treatment of *Conandron* varieties but also offers molecular support in lineage divergence of the Southeast-China-distributed *Conandron*. There are two lineage divergence events which occurred, leading to the genetic differentiation of *Conandron* into three distinct groups (Figure 4). The first lineage divergence occurred between mainland-Japan-distributed and Southeast-China- and Taiwan–Iriomote-distributed *Conandron*, which offer evidence in support of genetic distinctiveness of *C. ramondioides* var. *ramondioides* and *C. ramondioides* var. *taiwanensis*. The second lineage divergence event occurred between Taiwan–Iriomote-distributed and Southeast-China-distributed *Conandron* populations (Figure 4). Despite the Taiwan Strait being regarded as a dispersal route, allowing terrestrial species to migrate between the Asia continent and Taiwan during the glacial period, this is not the case for *Conandron.* Based on the lack of gene flow, Southeast-China- and Taiwan-distributed *Conandron* forming two distinct clades, and no shared haplotypes observed between Southeast-China- and Taiwan-distributed *Conandron*, one can postulate genetic differentiation has occurred between Southeast-China- and Taiwan-distributed *Conandron*.

Moreover, the modeled suitable habitat range provides another line of evidence in support of population differentiation between Southeast-China- and Taiwan-distributed *Conandron* (Figure 8). It is reported that the Taiwan Strait seafloor was exposed to air at least two times between 0.2–0.015 MYA [35], connecting the currently isolated Asian continent and Taiwan island. However, limited modeled suitable habitat range on the Taiwan Strait during the LGM period makes the Conandron migration between Southeast China and Taiwan through land bridge unlikely. The last line of evidence comes from *Conandron’s* biological features. The peduncle of *C. ramondioides* reaches about 9–15 cm above the ground in its forest habitat, and the stamen is connate at the basal position of the flower [36]. Furthermore, *C. ramondioides* lacks a hair-like appendage and may not be dispersed via animal. Based on these morphological features, overseas dispersal of *C. ramondioides* via wind or seed dispersers is unlikely. Thus, due to the lack of post-divergence gene flow, limited suitable habitat on the ECS during the LGM, and low dispersal ability, we postulate that a lineage diversification within *C. ramondioides* var. *taiwanensis* is likely.

### 3.4. Diversifying Selection May Contribute to Shape CrCYC1 Lineage Diversification

The reconstructed nuclear marker haplotype networks showed that *Conandron* haplotypes can be clustered into two or three distinct clades (Figure 2A–D). However, the distantly linked *CrCYC1* haplotype network arouses our attention. In addition, only *CrCYC1* haplotypes form three distinct clades by comparing with haplotype network topologies obtained from other nuclear markers. There are two possible evolutionary mechanisms behind this. The first one is the relatively high substitution rate of *CrCYC1* in *Conandron*. The second one is that *CrCYC1* may be affected by non-neutral evolutionary force in *Conandron*. The higher between-group genetic distance observed in *CrCYC1* (values of genetic distance ranging from 0.015 to 0.018) than those in *ITS* (values of genetic distance ranging from 0.003 to 0.004), provide evidence to support the first assumption (see Appendix A). The significant McDonald–Kreitman test results suggest that *CrCYC1* in *Conandron* is subjected to non-neutral evolution force. The relatively abundant non-synonymous substitutions (Dn) rather than synonymous substitutions (Ds) (e.g., 20 non-synonymous substitutions to 5 synonymous substitutions, when using *Cyrtandra* as outgroup, See Table 1), suggests diversifying selection could play a key role in shaping *CrCYC1* lineage diversification. This is because diversifying selection tends to maintain genetic diversity. Consequently, the *CYC* and its homologs are key genes in the establishment of floral symmetry with relaxed genetic constraints from the purifying selection [36,37,38,39,40]. Relaxing from genetic constraint allows genes to harbor in-frame substitutions. The relatively high haplotype and nucleotide diversity observed in *CrCYC1* compared with other nuclear markers (Appendix A) may provide evidence to support this assumption.

### 3.5. Divergent ITS Sequences Suggest That Conandron ITS Resulted from Incomplete Concerted Evolution

*C. ramondioides* is considered a diploid species [16]. We thus initially assumed that the nuclear haplotypes of the species should be hetero- or homozygotes, and that the sequenced ITS colonies should contain either hetero- or homozygotes. However, the number of sequenced *C. ramondioides* internal transcribed spacers (*ITS*) ranged from three to six, suggesting heterogeneity. *ITS* is one of the most popular molecular markers used for reconstructing plant phylogeny due to rapid concerted evolution within and among nuclear ribosomal DNA component subunits, fast substitution rate, and availability of universal primers [41,42,43]. Concerted evolution is a genetic exchange process among genes within a gene family, leading to the homogenization of different genes in a gene family [44]. However, high intra-individual polymorphism identified in different plant groups, such as *Lespedeza* [45], *Carapichea ipecacuanha* [46], *Pyrus* [47], and *Mammillaria* [48], suggests that incomplete concerted evolution of *ITS* in plants is common.

### 3.6. Lineage Discordance between cpDNA and nrDNA in C. ramondioides

When observing haplotype network topologies, the cpDNA haplotype network exhibits different topology compared with nrDNA haplotype networks (Figure 2A–D). There is a chloroplast haplotype observed in both Taiwan-distributed *Conandron* populations and the coastal-Fujian-distributed *Conandron* populations (Figure 2F, Appendix A). In addition, chloroplast haplotypes isolated from Southeast China formed three independent lineages in the cpDNA haplotype network (Figure 2F), which were never observed in other nrDNA haplotype networks (Figure 2A–D). Though incomplete lineage sorting hypothesis is applied to explain inconsistency between haplotype networks [49,50], it may not be the case here. The first observation is that the shared chloroplast haplotype (CP11) is not a widely distributed haplotype in the *Conandron* distribution range in East Asia, as would be expected because of the signal of incomplete lineage sorting. Second, the effective population size of cpDNA is four times smaller than that of nrDNA in a diploid species [51]. In other words, genetic drift should sweep out rare haplotypes four times faster in cpDNA than in nrDNA. If so, the reciprocal monophyly observed from the nrDNA pattern should be even more evident in cpDNA. However, it is not observed in the cpDNA haplotype network (Figure 2F). Furthermore, the genetic drift may not be the only force acting on the cpDNA of Southeast-China-distributed *Conandron*. The positive values obtained from the neutrality test suggested that cpDNA isolated from Southeast-China-distributed *Conandron* populations may experience diversifying selection (Table 2). The last potential reason which may contribute to the inconsistency between cpDNA and nrDNA here is the substitution rate difference between cpDNA and nrDNA. The measured parsimony informative sites of cpDNA are fewer than those measured from nrDNA dataset (0.024 from the cpDNA dataset; 0.059 from the AGT intron 1 dataset; 0.082 from the GroES1 dataset; and 0.055 from the LFY1 dataset), suggesting cpDNA evolving slower than nrDNA in *Conandron*. Hence, we conclude that the discrepancy between cpDNA and nrDNA in *C. ramondioides* var. *taiwanensis* is either caused by non-neutrally evolving cpDNA in the SE-China populations, slowly evolving cpDNA, or both.

In summary, the current disjunctive distribution pattern of *C. ramondioides* resulted from vicariance events. Three lines of evidence support this conclusion. Firstly, three groups corresponding to geographical regions were identified in our multi-loci phylogeny following restricted post-divergence gene flow. Secondly, there is a lack of potential past dispersal routes across an exposed ECS land bridge. Thirdly, we measured flower morphological traits (L/T) from the geographical regions, which revealed corresponding morphological groups. Our study represents a comprehensive approach to revealing the speciation process of East Asian endemic species by integrating information from molecular evolution, paleo-climate simulation, and morphological trait measurements.

## 4. Materials and Methods

### 4.1. Sample Collection and Genomic DNA Extraction

We sampled nine populations of *C. ramondioides* var. *ramondioides* and twelve populations of *C. ramondioides* var. *taiwanensis* (197 individuals in total), covering most of the species’ distribution range in East Asia (Figure 1). The exact localities are listed in Appendix A. Leaf samples were collected in the field and then dried with silica gel. Dried leaves were ground into powder for DNA extraction using a homogenizer with liquid nitrogen, following the cetyltrimethylammonium bromide (CTAB) procedure [52]. The DNA concentration for each sample was determined using a Nanodrop RNA/DNA Spectrophotometer (Thermo, Taipei, Taiwan).

### 4.2. Amplification of cpDNA Markers and Nuclear Markers

To evaluate the divergence patterns of *C. ramondioides* populations across East Asia, two cpDNA markers (*trn*L-*trn*F and *trn*H-*psb*A) and five nuclear markers (*CrCYC1*, *ITS, AGT* intron *1*, *GroES* intron *1*, and *LEAFY* intron *1*) were amplified from all sampled individuals. The first four markers (*trn*L-*trn*F, *trn*H-*psb*A, *CrCYC1,* and *ITS*) were sequenced using the Sanger sequencing method [53] in 2016. The primer pairs used for amplifying the *trn*L-*trn*F region were from Taberlet et al. [54]. The forward primer ‘*trn*H’ (CGC GCA TGG TGG ATT CAC AAT CC) and the reverse primer ‘*psb*A’ (GTTATGCATGAA CGT AAT GCT C) were used to amplify the *trn*H-*psb*A region [55]. The PCR reaction was made up to a total volume of 25 μL, containing 13 μL of 2× Ampliqon master mix Red (Ampliqon, Odense, Denmark), 0.25 μL of each primer at 2 μM, 10.5 μL of ddH_2_O, and 1 μL of template at 20 ng/μL. The PCR profile for cpDNA markers amplification was 35 cycles of 30 sec at 94 °C, 30 s at 56 °C, and 1 min at 72 °C, with a final extension of 5 min at 72 °C, after an initial denaturing for 3 min at 94 °C for both *trn*L-*trn*F and *trn*H-*psb*A. For nuclear markers, *CrCYC1* sequences were amplified using the forward primer “SPF” (5’-AGCAAGACATGCTTTCTGG-3’) and the reverse primer “SPR” (5’-GACATTGGATTCAATATGGTG-3’) [56], while the forward primer “ITS 5P” (5′-GGAAGGAGAAGTCGTAACAAGG-3′) and the reverse primer “ITS 8P” (5′-CACGCTTCTCCAGACTACA-3′) were used for amplifying the *ITS* region [57]. The PCR profile was 35 cycles of 30 s at 94 °C, 30 s at 58 °C, and 50 s at 72 °C, with a final extension of 10 min at 72 °C, after an initial denaturing for 3 min at 94 °C for *CrCYC1* and *ITS* amplification. The PCR reaction was made up to a total volume of 25 μL, containing 13 μL of 2× Ampliqon master mix Red, 0.25 μL of each primer at 2 μM, 10.5 μL of ddH_2_O, and 1 μL of template at 20 ng/μL. All PCR products were examined with 1.0% (*w*/*v*) agarose gel electrophoresis, and then the band of correct size was used for sequencing. The PCR products were sequenced using an ABI 3700 automatic sequencer (Genomics, New Taipei City, Taiwan). All sequences were examined firstly with a Sequence Scanner v1.0 and then aligned for further analysis. To examine within-individual homogeneity of *ITS*, we amplified nine individuals (three from Japan, three from Southeast China, and three from Taiwan) using *ITS* primer pairs with a proofreading KAPA HiFi PCR kit (KAPA Biosystems, Potters Bar, UK). These nine PCR products were cloned using a pGEM-T Easy Vector kit (Promega, Madison, WI, USA), following the manufacturer’s protocols. Finally, five to six clones of each PCR product were sequenced. Sequences were deposited in GenBank under the following Accession Numbers: *CrCYC1* (OP631034-OP631413), *ITS* (OP578698-OP579075), *trn*L-*trn*F (OP689756-OP689952) and *trn*H-*psb*A (OP689953-OP690149).

We developed specific single-copy nuclear intron markers for *C. ramondioides* in 2018 [58]. Due to the species’ high genetic variability, single-copy nuclear intron markers are preferable to non-coding cpDNA markers [59]. Out of eleven potential single-copy markers, three (AGT intron 1, GroES intron 1, and LEAFY intron 1) were selected for amplicon library preparation. These three markers generated a single band on the gel electrophoresis. Amplicon library preparations of these three loci were conducted following the 16S metagenomic sequencing library preparation guide provided by Illumina Technology (San Diego, CA, USA). The amplicon library was then sequenced on the Illumina MiSeq sequencing system. Paired-end Illumina reads (2 × 300 bp) were imported into CLC Bio Genomics workbench 7.0.4 (QIAGEN Bioinformatics, Venlo, Netherlands) and merged to derive overlapping pairs. Only reads with over 100 counts were kept for further analysis. Sequences were deposited in GenBank under the following Accession Numbers: AGT intron 1 (OP598596-P598799), GroES intron 1 (OP631506-OP631663 and MH311300-MH311413), and LEAFY intron 1 (OP631414-OP631505 and MH237973-238072).

### 4.3. Genetic Structure of Populations and Geographical Structure Analysis

Sequences of all six markers (ATG2 intron 1, GroES intron 1, LFY intron 1, *CrCYC1*, *ITS,* and concatenated cpDNA) were aligned with the MUSCLE algorithm [60] implemented in MEGA 6 [61] and then manually edited using BioEdit v. 7.2.5 [62]. We estimated nucleotide diversity (π) and haplotype diversity (Hd) using DnaSP ver 5.0 [63] for each marker for each population, assigned region, and at the species level. When measuring the genetic diversity of populations on both sides of the ECS, we assigned the Honshu and Shikoku populations as the North group and the Taiwan, Iriomote, and SE-China populations as the South group (see Appendix A). The Hd and π values of East Asian herbaceous plants have a range of 0.61–0.9 and 0.0001–0.003, respectively [14,15,24].

The ECS is considered a barrier inhibiting gene flow for shrubs or herbs [12,15,24]. Thus, *Conandron* populations distributed on the main islands of Japan (Honshu and Shikoku), Taiwan, and Southeast China were classified into these separate geographical regions for examination of the genetic structure. Iriomote populations were assigned to the Taiwan group since they are geographically close to Taiwan. An analysis of molecular variance (AMOVA) of our genetic data was performed using the program Arlequin ver 3.5 [64]. *F*-statistics were calculated to estimate the proportion of gene variability found among populations (*F*st), among the population within groups (*F*sc), and among groups (*F*ct) of our datasets. If the ECS serves as a barrier, we would expect the highest genetic variation to be attributed to the group level. The statistical significance of all estimated fixation indices was tested with 10,000 permutations [65].

To assess the genetic relationships of sampled *C. ramondioides* individuals for each tested marker, we constructed a statistical parsimony network with a 95% connection limit using the R package “Haplotypes” [66]. Indels were coded using the simple indel coding method, which treated indels with different start and end positions as separate characters [67].

### 4.4. Inferring Lineage Divergence of C. ramondioides Populations from Nuclear Markers

To rebuild the relationship among *Conandron* populations, *BEAST with multi-locus species tree examination in the BEAST v2.4.7 program [68,69] was used to estimate gene trees. Each isolated sequence could be assigned into a group along with the collection site. Each collection site was treated as an operational taxonomic unit (OTU) in species tree analyses. The program jModeltest v2.1.3 was used to estimate the substitution model of each locus [70], the HKY model was set for the LFY and GroES1 dataset, and HKY+G for the ATG2 intron 1 dataset. Considering the low rate of discrepancy anticipated at the intraspecific level, a strict clock model was selected for all three loci [71]. For the clock rate prior, a mean and initial value of 0.0037 per substitutions/site/million years and a standard deviation of 2.55 (95% interquartile range of 0.012–0.0027) were set under a lognormal prior distribution of the LFY dataset (Appendix A). The substitution rates were also determined by a mean value and initial value of 0.0037 per substitutions/site/million years with more broadly distributed lognormal prior distributions (ATG2 intron 1 and GroES intron 1: 95% interquartile range of 0.187–0.000074) to cover a wide range of substitution rate of these two loci. The Yule tree prior and a piecewise linear root population size model for species tree priors were used. Markov chain Monte Carlo (MCMC) runs were set for 50 million generations with sampling every 5000 generations and discarding the first 10% as burn-in. Three independent MCMC chains with these settings were performed. The log and species tree files from the replicates were combined into single-log and single-tree files using the logCombiner program in BEAST v2.4.7 [68,69]. The convergence level of these three MCMC chains could be visualized and checked in the program Tracer v1.6.0 [72].

The grouping pattern was inferred from the multi-loci species tree; we further reconstructed the population differentiation process by gene flow with an isolation with migration model, which was implemented in IMa2 [73]. An infinite site model was set for each of the three nuclear markers. Truncated uniform distribution priors of all six parameters of the IMa2 were set according to the author’s recommendations [73]. To cover a broad range of these scaled parameters, including lineage splitting time, effective population size of both extant and ancestral populations, and the gene flow among assigned groups [3], the upper bond of these parameters were set with the theta estimated from the Taiwan–Iriomote group, which had the highest theta of the three *Conandron* groups. The geometric mean of Watterson’s theta [74] per sequence over three markers was 3.2. The mutation rate of LFY (per locus with 3.7 × 10^−9^ × 393 bp for the LFY marker) was used as a reference for the other two loci to rescale the IMa2 parameters.

Furthermore, the same upper bound settings were applied to our pairwise dataset of the three *Conandron* groups. Divergence time estimated from a pairwise analysis will show splitting order among groups [75]; therefore, the results confirm the lineage divergence process inferred from both multi-population IMa2 and *BEAST2. Finally, 40 Metropolis-coupling MCMC (MCMCMC) chains with heating parameters of -ha0.96 and -hb0.85 were set for all pairs. The first 10^5^ iterations of pairwise group pairs were discarded as burn-in, followed by 10^6^ iterations for sampling (100 iterations per sample). To confirm whether our analysis exhibited convergence, three independent runs were performed. To determine the statistical significance of gene flow between groups, the Likelihood ratio tests (LRT) were computed [76].

### 4.5. Population Demographical History

To evaluate whether the molecular markers conformed with the expectations of neutrality theory, both Tajima’s D [77] and Fu’s Fs [78] were applied using DnaSP5.0 [32], based on 1000 coalescent simulations. The two statistics are expected to be nearly zero or zero for a stable population, but significant negative values indicate a sudden expansion in population size, while significant positive values suggest the occurrence of processes such as a population subdivision or recent population bottlenecks. As a transcription factor, selection could affect *CrCYC1* at the codon level. Therefore, the *CrCYC1* dataset was subjected to further examination of deviation from neutral evolution using the McDonald–Kreitman test [79] as implemented in the McDonald–Kreitman test online web interface [80].

We reconstructed the effective population size change in *Conandron* groups (including Honshu–Shikoku, Taiwan–Iriomote, and Southeast China groups) through time by assessing whether the species had undertaken population expansion or decline during cold periods (e.g., last glacial maximum). An extended Bayesian skyline plot (EBSP) implemented in BEAST v2.4.7 was used, and EBSP uses coalescent-based models to estimate posterior probabilities of effective population size change through time [81]. Substitution models for each dataset and molecular clock prior were set as for the *BEAST analysis. The “coalescent: constant size” tree prior was set for each group in three EBSP runs. Each run comprised 50 million generations with sampling every 5000 steps, of which the first 10% were discarded as burn-in. We integrated individual information from the three log files and the corresponding three gene trees into a single file using the logCombiner program in BEAST v2.4.7. Skyline plots were generated in Microsoft EXCEL. Bayes factors (BF) of each group were then measured to evaluate statistical significance of each EBSP. A BF greater than 3 suggested significant population size change through time.

### 4.6. Simulation of the Least-Cost Path to Visualize Potential Dispersal Routes of Conandron Populations during the Last Glacial Maximum in East Asia

To estimate the least-cost path between populations, we first generated a resistance map by using a reciprocal of a suitable map. Resistance-to-suitability mapping is simply a monotonic transformation in which locations with higher suitability exhibit lower resistance. Therefore, the potential suitable habitat range of *Conandron* was modeled using the maximum entropy method [82,83]. Firstly, available specimen occurrence data were acquired from GBIF (http://www.gbif.org/, accessed on 1 December 2017), and 19 BIOCLIM variables from the WorldClim data set [84], including “current” and “last glacial maximum”, were used at a 2.5-mile spatial resolution (~5 km). Secondly, potentially suitable habitat ranges for the “current occurrence” of *Conandron* were modeled and then projected to the “last glacial maximum occurrence” layers to show potential suitable habitat ranges of *Conandron* during the LGM period.

Finally, the pairwise least-cost path was measured among 21 populations from the resistance map using the R package “gdistance” [85]. The resistance map was first converted to a raster format composed of cells. In “gdistance”, the calculation of routes starts with connecting cell centers to each other, which become nodes on the resistance map. “gdistance” implements three cell connection methods: von Neumann, Moore, and king’s and knight’s move. In the von Neumann method, cells connect orthogonally to their four neighbors. In the Moore method, cells connect to their eight nearest neighbors orthogonally and diagonally. In the king’s and knight’s moves method, cells connect to their sixteen neighbors vertically over one square and orthogonally, diagonally, and horizontally across two squares. To explore all possible routes, all three connecting methods were tested.

### 4.7. Identification of Cryptic Species of C. ramondioides via Floral Morphology Measurements

Corolla morphology (lobe length to tube length ratio) has been proposed as a key trait to distinguish *C. ramondioides* populations distributed across Japan from those in Taiwan and Iriomote [16]. Therefore, the available *C. ramondioides* corolla was measured following the method of Kokubugata and Peng [24].

## Figures and Tables

**Figure 1 ijms-23-14932-f001:**
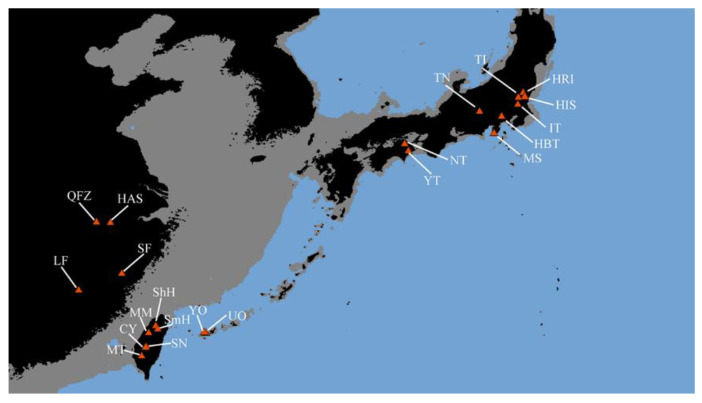
Locations of *C. ramondioides* sample collection. Samples were marked with abbreviations corresponding to those in Appendix A. Shaded sea areas indicate continental shelves that might have been exposed during the last glacial maximum.

**Figure 2 ijms-23-14932-f002:**
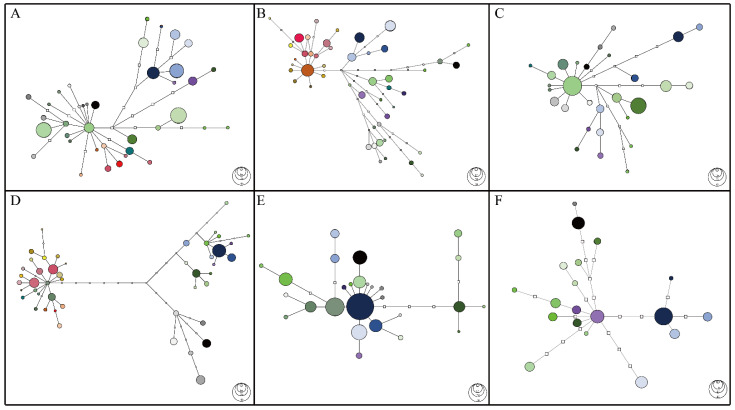
Statistical parsimony networks reconstructed from the six loci isolated from *C. ramondioides*. Hollow squares indicate hypothetical (i.e., extinct or not sampled) haplotypes; one mutational step is represented as a line connecting to each haplotype. The size of colored circle corresponds to haplotype frequency. (**A**). ATG2 intron 1, (**B**). GroES intron 1, (**C**). LFY intron 1, (**D**). *CrCYC1*, (**E**). *ITS*, and (**F**). cpDNA (*trn*L-F + *trn*H-*psb*A). Color codes are listed in Appendix A. Haplotype and ribotype frequency maps are shown in Appendix A.

**Figure 3 ijms-23-14932-f003:**
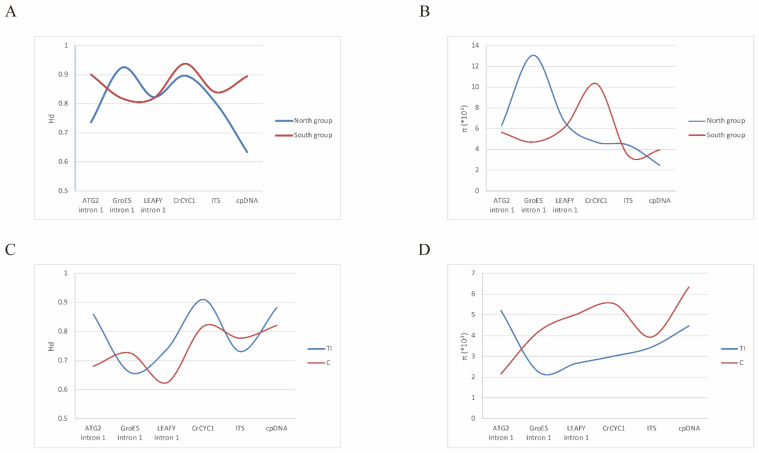
Genetic index values were measured from nuclear and cpDNA markers. (**A**,**B**): Hd and π were measured from five nuclear markers (ATG2 intron 1, GroES intron 1, LEAFY intron 1, *CrCYC1,* and *ITS*) and cpDNA of the North group (Honshu and Shikoku) and the South group (Southeast China, Taiwan, and Iriomote). (**C**,**D**): Hd and π measured from the five nuclear markers and cpDNA of the TI group (Taiwan and Iriomote) and the C group (Southeast China).

**Figure 4 ijms-23-14932-f004:**
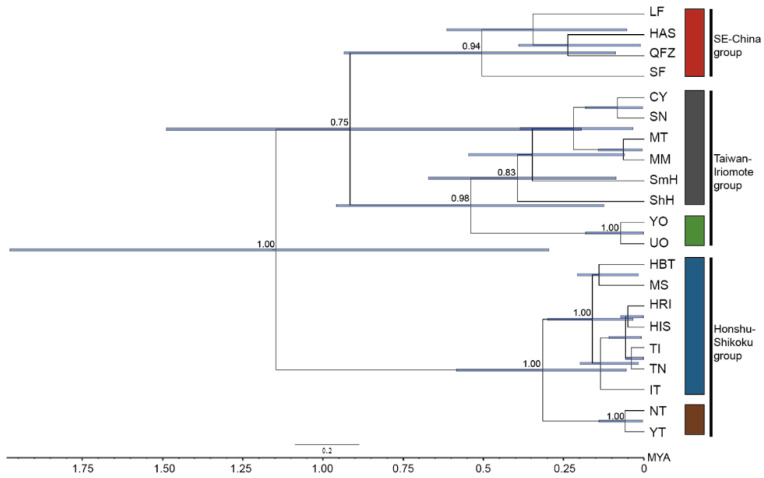
Phylogeny of sampled *C. ramondioides* populations reconstructed by *BEAST2. The phylogenetic tree was reconstructed from the 3nrDNA dataset. Numbers above nodes represent posterior probability. Colored squares represent the geographic region.

**Figure 5 ijms-23-14932-f005:**
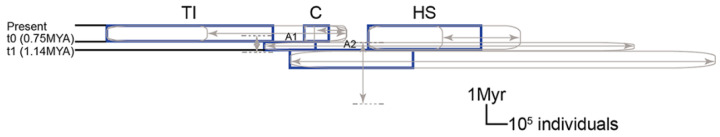
The reconstructed lineage divergence process inferred from a multi-population IMa2 analysis of *C. ramondioides*. TI denotes sampled populations distributed in Taiwan and Iriomote, C denotes sampled populations distributed in Southeast China, and HS denotes sampled populations distributed in Honshu and Shikoku. In the cartoon plot, effective population size of each clade and lineage divergence time among groups are indicated by a box and horizontal lines, respectively. Blue boxes and blue horizontal lines are scaled to denote the mean measured parameters. Gray boxes and lines, including double arrows and dashed lines, are scaled to present the 95% confidence intervals of the corresponding parameters. A1 denotes the ancestor population of C and TI. A2 denotes the ancestor population of C, TI and HS.

**Figure 6 ijms-23-14932-f006:**
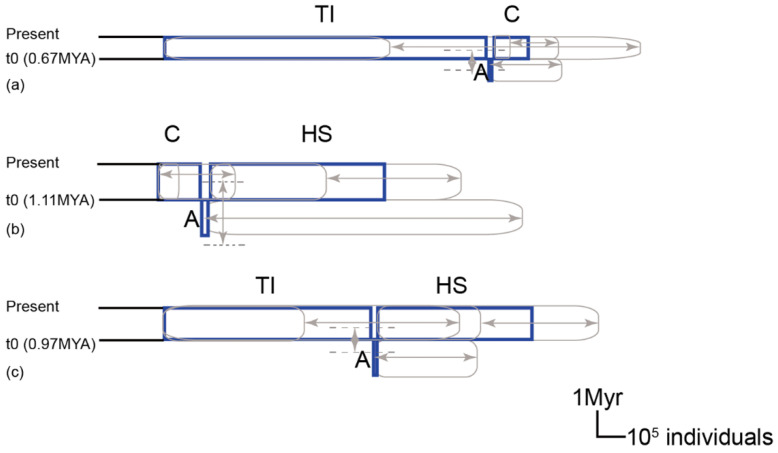
Population divergence process among assigned *C. ramondioides* groups inferred from a pairwise IMa2 analysis. The group codes, figure labels, and figure format correspond to Figure 5. Scale bars for 100,000 individuals and 1 million years apply to all six plots. (**a**) The estimated divergence process between TI and C group. (**b**) The estimated divergence process between C and HS group. (**c**) The estimated divergence process between TI and HS group.

**Figure 7 ijms-23-14932-f007:**
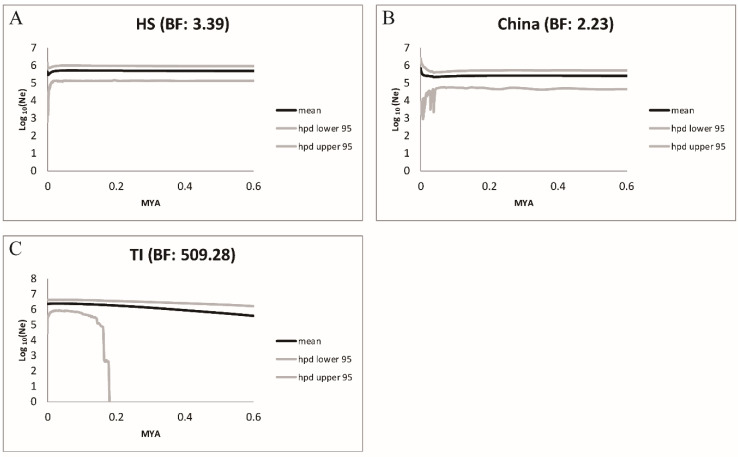
The effective population size changes over time of *C. ramondioides* groups. (**A**). HS: Honshu–Shikoku group, (**B**). China: Southeast China group, (**C**). TI: Taiwan–Iriomote group.

**Figure 8 ijms-23-14932-f008:**
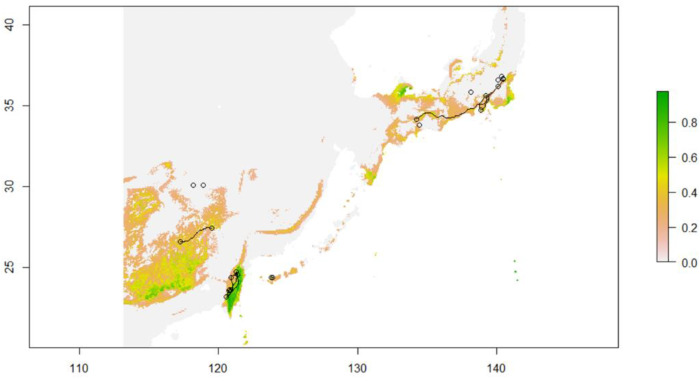
Modeled potential suitable habitat range and least-cost path of *C. ramondioides* during the last glacial maximum (LGM). Gray: land configuration during the LGM. Hollow circles: populations sampled during this study. Color bar: green to white denotes suitable to unsuitable habitat for *Conandron* during the LGM.

**Figure 9 ijms-23-14932-f009:**
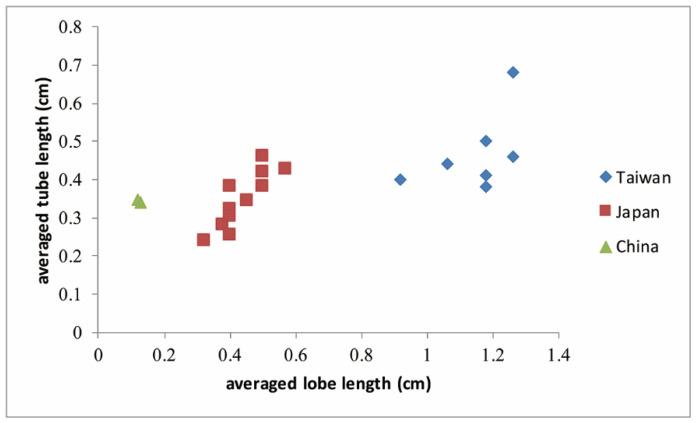
Corolla tube versus lobe length measured from *C. ramondioides* flowers distributed in China, Taiwan, or Japan.

**Table 1 ijms-23-14932-t001:** McDonald–Kreitman tests of *CrCYC1C* haplotypes.

	*Cyrtandra* *		*Hemiboea* *		*Oreocharis* *	
	Polymorphism	Divergence	Polymorphism	Divergence	Polymorphism	Divergence
Synonymous (dS)	5	24	9	20	8	11
Non-synonymous (dN)	20	24	41	30	16	21
Total	25	48	50	50	24	32
χ^2^ value	6.11		5.876		0.904	
*p*-value	**0.013**		**0.015**		0.34	

* These three Gesneriaceae species are used as outgroups.

**Table 2 ijms-23-14932-t002:** Neutrality tests of four molecular markers estimated from the China populations.

	Tajima’s D	Fu’s Fs
cpDNA	2.9 (*p* < 0.01)	7.86 (*p* < 0.01)
ATG2 intron 1	−0.15 (*p* > 0.1)	0.05 (*p* > 0.1)
GroES intron 1	1.63 (*p* > 0.1)	2.04 (*p* > 0.1)
LFY intron 1	1.3 (*p* > 0.1)	3.34 (*p* > 0.1)

**Table 3 ijms-23-14932-t003:** Pairwise Fst estimates from four loci of three identified *C. ramondioides* groups.

	Group	Honshu+Shikoku	Taiwan+Iriomote
	Honshu+Shikoku	-	
(a) ATG2 intron 1	Taiwan+Iriomote	0.571 ***	-
	Southeast China	0.748 ***	0.691 ***
	Honshu+Shikoku	-	
(b) GroES intron 1	Taiwan+Iriomote	0.449 ***	-
	Southeast China	0.298 ***	0.513 ***
	Honshu+Shikoku	-	
(c) LFY intron 1	Taiwan+Iriomote	0.472 ***	-
	Southeast China	0.674 ***	0.848 ***
	Honshu+Shikoku	-	
(d) cpDNA	Taiwan+Iriomote	0.295 ***	-
	SE-China	0.32 ***	0.269 ***

*** *p* < 0.001.

**Table 4 ijms-23-14932-t004:** Hierarchical analysis of molecular variance (AMOVA) in *Conandron* obtained from sequences of three loci.

		ATG2 intron 1				
3 groups						
Source of variation	d.f.	Sum of squares	Variance components	Percentage variation	Fixation indices	*p*
Among groups	2	100.635	0.771	39.61	ΦCT = 0.4	0.003
Among populations within groups	7	75.023	0.721	37.04	ΦSC = 0.61	<0.0001
Within populations	144	65.443	0.454	23.34	ΦST = 0.767	<0.0001
Total	153	153.000	1.947			
		GroES intron 1				
Source of variation	d.f.	Sum of squares	Variance components	Percentage variation	Fixation indices	*p*
Among groups	2	154.583	0.844	36.750	ΦCT = 0.367	<0.0001
Among populations within groups	12	109.228	0.541	23.550	ΦSC = 0.372	<0.0001
Within populations	226	205.067	0.911	39.700	ΦST = 0.601	<0.0001
Total	239	468.367	2.290			
		LFY intron 1				
Source of variation	d.f.	Sum of squares	Variance components	Percentage variation	Fixation indices	*p*
Among groups	2	77.365	0.839	47.810	ΦCT = 0.478	0.002
Among populations within groups	7	64.485	0.657	37.410	ΦSC = 0.717	<0.0001
Within populations	124	32.166	0.259	14.780	ΦST = 0.852	<0.0001
Total	133	172.769	1.741			

## Data Availability

All sequences data are submitted to NCBI.

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
