# Peer review of "Allopatric Lineage Divergence of the East Asian Endemic Herb Conandron ramondioides Inferred from Low-Copy Nuclear and Plastid Markers"

_ijms, 2022, doi:10.3390/ijms232314932_

Round 1
Reviewer 1 Report
The submitted manuscript contributes substantially towards a better understanding of evolutionary history of an East Asian monotypic genus, Conandron. The data presented are scientifically sound and substantiated by original data. The manuscript is suitable for publication, after having undergo some revisions:
Line 92: The example of ITS as a single gene is not a good one.
Lines 150-155: When referring to ITS, it is better to use the ribotype instead of haplotype.
Lines 156-164: The study shows that nucleotide additivity occurs in the ITS sequences. This feature needs to be discussed in more detail. What is most likely is that additive nucleotides reflect hybridization or a recent divergence of the isolated groups?
Lines 198-206: Additional clarifications are needed here. Is there evidence that recombination between different copies also contributed to the diversity of ITS?
Lines 542-545: Authors conclude incomplete concerted evolution of ITS in plants is common. At the same time, the authors cite works in which either acts of hybridization or non-functional copies were identified. Please specify what processes leading to the diversification of ITS could take place in the case of Conandron.
Lines 475-486: Figure 9 shows that, according to the characteristics of the flower, Conandron ramondioides can be divided into three groups. However, only two varieties are mentioned in the text. What is the reason for these inconsistencies? If new data were obtained that contribute to the description of another variety, this should be formulated more clearly.
Lines 504-506: The authors report that the closest relative of Conandron is Lysionotus pauciflorus. However, in the work of Möller et al. (Plant Syst Evol 2011, 292: 223–248), other plant species were identified as closely related.
Methods: How were the parsimony networks reconstructed, given nucleotide additivity found in the ITS sequences? Also, please specify if there was a problem with possible paralogous sequences of low copy genes and how it was overcome?
Figure 2: I encourage the authors to redo this figure. I propose to color the haplotype groups, not the geographic location of the samples, and also present distribution map of haplotype groups in the populations of Conandron ramondioides.
Author Response
The submitted manuscript contributes substantially towards a better understanding of evolutionary history of an East Asian monotypic genus, Conandron. The data presented are scientifically sound and substantiated by original data. The manuscript is suitable for publication, after having undergo some revisions:
Line 92: The example of ITS as a single gene is not a good one.
Reply: Thanks for pointing out this example. We have changed the example of single copy gene from ITS to LEAFY. Since the LEAFY gene has been known as a single copy gene in most angiosperm (Baum et al. 2005. Molecular evolution of the transcription factor LEAFY in Brassicaceae. Molecular Phylogenetics and Evolution. 37(1) :1-14).
Lines 150-155: When referring to ITS, it is better to use the ribotype instead of haplotype.
Reply: Thanks for your suggestion. We accept this suggestion and correct ITS “haplotype” to ITS “ribotype” accordingly.
Lines 156-164: The study shows that nucleotide additivity occurs in the ITS sequences. This feature needs to be discussed in more detail. What is most likely is that additive nucleotides reflect hybridization or a recent divergence of the isolated groups?
Reply: Thanks for your suggestion. By only examining the ribotypes of ITS, one can postulate recent divergence event or hybridization occurred among isolated groups due to shared ribotypes. However, if this is true, we expect we should observe shared haplotypes among groups from other nuclear markers. In this study, five nuclear markers are amplified and analyzed. Only ribotypes of ITS exhibit shared ribotypes among isolated groups. Therefore, it is not confident to assume recent divergence event or hybridization occurred in Conandron, relying on only ITS ribotypes.
Lines 198-206: Additional clarifications are needed here. Is there evidence that recombination between different copies also contributed to the diversity of ITS?
Reply: Thanks for pointing out the potential role of recombination. We detect recombination event occurred within ITS (site ranges from 260 to 306) by applying DnaSP v5. Therefore, we add a description of recombination in Conandron to highlight its contribution (lines 205-207).
Lines 475-486: Figure 9 shows that, according to the characteristics of the flower, Conandron ramondioides can be divided into three groups. However, only two varieties are mentioned in the text. What is the reason for these inconsistencies? If new data were obtained that contribute to the description of another variety, this should be formulated more clearly.
Reply: To date, the Conandron is classified into two varieties. Flower trait of southeast China distributed Conandron is not measured before. It is not confident to propose a “new” variant with only few evidence (Only two flowers are collected during our field trip in southeast China). However, our finding provides an opportunity for further exploring Conandron floral morphology of those distributed in China. With examining more samples from China, we may be able to propose a new variety in China in the future.
Lines 504-506: The authors report that the closest relative of Conandron is Lysionotus pauciflorus. However, in the work of Möller et al. (Plant Syst Evol 2011, 292: 223–248), other plant species were identified as closely related.
Reply: Thanks for pointing out this report. After checking the publication, the genus Ridleyandra, is the closest relative to Conandron. We agree with your point and remove the adjective “closely” in our manuscript (See line 507).
Lines 542-545: Authors conclude incomplete concerted evolution of ITS in plants is common. At the same time, the authors cite works in which either acts of hybridization or non-functional copies were identified. Please specify what processes leading to the diversification of ITS could take place in the case of Conandron.
Reply: Thanks for your suggestion. Here we aim to say the “incomplete concerted evolution of ITS” phenomenon is observed in diverse plant species or lineages. No matter hybridization or non-functional copies occurred or identified, the incomplete concerted evolution of ITS is observed. In Conandron, considering the haplotypes obtaining from other nuclear markers, the hybridization leading to incomplete concerted evolution of ITS is unlikely. We may postulate the recombination process is not completed in Conandron, which may lead to incomplete concerted evolution of Conandron ITS. (Lines 543-547).
Methods: How were the parsimony networks reconstructed, given nucleotide additivity found in the ITS sequences? Also, please specify if there was a problem with possible paralogous sequences of low copy genes and how it was overcome?
Reply:
- To conducted the ITS ribotype, we first reconstruct ITS ribotypes by applying program PHASE (Stephens et al. 2001. A New Statistical Method for Haplotype Reconstruction from Population Data. American Journal of Human Genetics. 68, 978-989.). Reconstructed ITS ribotypes were subjected to R package “Haplotypes” to reconstruct ribotype network. We noticed the multi-copy issue of ITS, thus exclude the ITS dataset from further analysis.
- We do not encounter the paralogous issue you mentioned.
Figure 2: I encourage the authors to redo this figure. I propose to color the haplotype groups, not the geographic location of the samples, and also present distribution map of haplotype groups in the populations of Conandron ramondioides.
Reply: Thanks for your suggestion. However, we aim to show the regional specific distribution of Conandron haplotypes/ribotypes. Therefore, we colored the haplotypes with geographic distribution of Conandron samples. In addition, we have tried to put all haplotypes (or ribotypes) on the map previously, but many colored haplotypes might be confusion to readers.
Reviewer 2 Report
The authors of the manuscript, “ijms-2013506” examined the origin and divergence processes of an East Asian endemic ornamental plant (C. ramondioides) through reconstructing the evolutionary and population demography history of C. ramondioides. They showed that its populations should be classified into three clades corresponding to geographical regions which are: Japan (Honshu+Shikoku), Taiwan-Iriomote, and Southeast China. They further highlighted the critical influence of species' biological characteristics on shaping lineage diversification of East Asian relic herb species during climate oscillations since the Quaternary. In my opinion, this research will make valuable contributions to the literature and science, and can be considered for publication in the Journal.
Nonetheless, the authors should pay attention to the following minor revisions
Point 1: Include recent references in the research.
Point 2: Polish the English usage and content delivery of some parts of the manuscript, for example; Line 45-6, and 181-2 can be rephrased.
Point 3: Provide clear Figures of higher resolution, in both the manuscript and the supplementary file. Additionally, the Supplementary Table 1 can be re-arrangement, it appears chaotic.
Author Response
Reviewer2
Comments and Suggestions for Authors
The authors of the manuscript, “ijms-2013506” examined the origin and divergence processes of an East Asian endemic ornamental plant (C. ramondioides) through reconstructing the evolutionary and population demography history of C. ramondioides. They showed that its populations should be classified into three clades corresponding to geographical regions which are: Japan (Honshu+Shikoku), Taiwan-Iriomote, and Southeast China. They further highlighted the critical influence of species' biological characteristics on shaping lineage diversification of East Asian relic herb species during climate oscillations since the Quaternary. In my opinion, this research will make valuable contributions to the literature and science, and can be considered for publication in the Journal.
Nonetheless, the authors should pay attention to the following minor revisions
Point 1: Include recent references in the research.
Reply: Thanks for pointing out this. A recent published paper “Wei, X.-P. and Zhang, X.-C. (2022), Phylogeography of the widespread fern Lemmaphyllum in East Asia: species differentiation and population dynamics in response to change in climate and geography. J. Syst. Evol., 60: 411-432.” is included. It is listed in Line 466-467.
Point 2: Polish the English usage and content delivery of some parts of the manuscript, for example; Line 45-6, and 181-2 can be rephrased.
Reply: Thanks for your suggestion. The population differentiation process is now rephrased (line 46). The MK test description is condensed for clearly reading (lines 181-182).
Point 3: Provide clear Figures of higher resolution, in both the manuscript and the supplementary file. Additionally, the Supplementary Table 1 can be re-arrangement, it appears chaotic.
Reply: The figures are provided with 300dpi following IJMS criteria. We have re-rearranged the supplementary Table 1.
Round 2
Reviewer 1 Report
I still have some major concerns: some of these I have mentioned previously, others are new.
Figure 2: I encourage the authors to redo this figure. I propose to color the haplotype groups, not the geographic location of the samples, and also present distribution map of haplotype groups in the populations of Conandron ramondioides.
Methods: How were the parsimony networks reconstructed, given nucleotide additivity found in the ITS sequences?
Lines 158 - 166: I am not sure that genetic terms such as allele and homo- or heterozygote can be used in relation to ITS, as there may be differences between copies located on the same chromosome. Therefore, I urge the authors to avoid genetic terms in this case and use more neutral ones.
Lines 507-508: It remains unclear why the authors compare Conandron with Lysionotus pauciflorus, especially if they are not closely related.
Author Response
Figure 2: I encourage the authors to redo this figure. I propose to color the haplotype groups, not the geographic location of the samples, and also present distribution map of haplotype groups in the populations of Conandron ramondioides.
Reply: Following your suggestions, the Figure 2 is updated (line: 147-148). For clarity, frequency maps of six loci used here are present separately in the supplementary Figure2. Due to the substitution rate variance, it is not easy to have a concise conclusion when presenting frequency map of all six molecular markers at the same time. Therefore, the frequency maps are presented in supplementary Figure 2.
Methods: How were the parsimony networks reconstructed, given nucleotide additivity found in the ITS sequences?
Reply: To show the relationship among isolated ITS ribotypes, the ribotype network is reconstructed. We are awareness of the nucleotide additivity found in our dataset, thus the ITS dataset is excluded for further analysis. In addition, we tend to show the process how we explain topology inconsistency among isolated molecular markers, instead of ignore it. Therefore, we apply the tools and program (R package “Haplotypes”) to show the relationship among ITS ribotypes.
Lines 158 - 166: I am not sure that genetic terms such as allele and homo- or heterozygote can be used in relation to ITS, as there may be differences between copies located on the same chromosome. Therefore, I urge the authors to avoid genetic terms in this case and use more neutral ones.
Reply: Thanks for your suggestion. The description is removed from the paragraph (Description in line164-165 is now removed).
Lines 507-508: It remains unclear why the authors compare Conandron with Lysionotus pauciflorus, especially if they are not closely related.
Reply: The description is now rephrased. Instead of comparing to L. pauciflorus, now the potential function of hair like appendage is not existed instead (line502-503).